# Physical Examination for Endocrine Diseases: Does It Still Play a Role?

**DOI:** 10.3390/jcm11092598

**Published:** 2022-05-05

**Authors:** Andrea Crafa, Rosita A. Condorelli, Rossella Cannarella, Antonio Aversa, Aldo E. Calogero, Sandro La Vignera

**Affiliations:** 1Department of Clinical and Experimental Medicine, University of Catania, 95123 Catania, Italy; crafa.andrea@outlook.it (A.C.); rosita.condorelli@unict.it (R.A.C.); rossella.cannarella@phd.unict.it (R.C.); acaloger@unict.it (A.E.C.); 2Department of Experimental and Clinical Medicine, University Magna Graecia of Catanzaro, 88100 Catanzaro, Italy; aversa@unicz.it

**Keywords:** physical signs, endocrinology, endocrine diseases, semeiotics

## Abstract

A physical examination represents a fundamental step in diagnosing diseases. Due to the role that hormones play in the regulation of numerous biological processes in various organs and systems, endocrine diseases cause a variety of clinical manifestations that can be easily identified with a careful physical examination and can guide the clinician to specific diagnoses. Furthermore, the presence of specific clinical signs in various endocrine-metabolic diseases can predict the risk of developing comorbidities and serious adverse events. In this article, we present some of the main clinical signs of endocrine-metabolic diseases and the risk of comorbidities, summarizing the pathogenetic mechanisms that lead to their formation. The aim is to highlight how the identification of these specific signs can reduce the number of dynamic tests and the costs necessary to reach the diagnosis and allow the early identification of any complications associated with these diseases, improving the clinical management of affected patients.

## 1. Introduction

Physical examinations represent a milestone in the clinical management of a patient, guiding the clinician towards better use of diagnostic tests and a reduction in the demand for unnecessary laboratory and instrumental tests [1]. Does this also apply to Endocrinology? We present an overview of the main clinical signs encountered during the common clinical practice by endocrinologists, highlighting their diagnostic and, in some cases, even prognostic roles.

## 2. Main Palpatory Signs Associated with Endocrine Diseases

### 2.1. Palpatory Signs in Thyroid Diseases

Physical examination certainly plays a key role in the diagnosis of thyroid diseases. The presence of goiter, suitable correlated to the patient’s symptoms, allows us to suspect the presence of thyroid functional abnormalities (hypothyroidism or thyrotoxicosis) which can then be confirmed by hormonal tests. A palpable, painful goiter in patients with fever and initial symptoms of thyrotoxicosis suggests the presence of subacute thyroiditis. Adequate palpation can also identify the presence of nodules although it is not possible to distinguish a cystic nodule from a solid one by simple palpation. However, palpation of a very hard and firm nodule, possibly associated with later cervical lymphadenopathy, increases the suspicion of neoplastic disease. The presence of several palpatory nodules associated with symptoms of thyrotoxicosis suggests the presence of a toxic multinodular goiter, while a single nodule with symptoms of thyrotoxicosis suggests the presence of Plummer’s adenoma. Finally, the presence of Pemberton’s sign positivity, characterized by the appearance of facial congestion and respiratory distress when the patient is asked to raise his arms, may be indicative of an endothoracic goiter that compresses the veins and trachea due to the narrowing of the superior thoracic entrance caused by the maneuver. Therefore, thyroid palpation cannot be neglected in the management of patients with symptoms referable to thyroid dysfunction [2].

Palpation of the neck may also reveal the presence of thyroglossal duct cysts that present as a painless, mobile mass of the midline of the neck, 75% of cases located inferior to the hyoid bone and showing movement with protrusion of the tongue [3].

Another important palpatory sign associated with thyroid dysfunction is pretibial myxedema. Its presence increases the suspicion of hyperthyroidism and, in particular, of Graves’s disease (GD). This form of non-pulsating edema is not limited to the pretibial region and can also appear in other locations such as forearms, shoulders, arms, palms, upper back, and neck. Therefore, it can also be referred to as thyroid dermopathy. It has a prevalence of 4.3% of patients with GD rising to 15% in patients who also have ophthalmopathy. It is characterized by the presence of bilateral, asymmetrical, non-puncturing, painless, and erythematous nodules and plaques due to the deposition of mucopolysaccharides, in particular, hyaluronic acid in the papillary and reticular dermis with extension into the subcutis [4].

### 2.2. Palpatory Signs in Scrotal Diseases

For the evaluation of the infertile male, one of the signs that most correlates with sperm function is the low testicular volume (<12 mL). Most of the testicular volume depends on the proliferation of germ cells and the enlargement of seminiferous tubules during puberty. Therefore, spermatogenic impairment can be suspected in the presence of low testicular volume. The lower the testicular volume, the greater the damage to the germ cell component [5]. Furthermore, particularly small testes (bilateral testicular volume ≤ 5 mL) with increased consistency suggest the presence of Klinefelter syndrome, to be confirmed by sperm analysis, the hormonal profile of hypergonadotropic hypogonadism, and genetic tests [6]. On the other hand, in the case of children who come to the endocrinologist’s attention due to growth retardation and developmental delay of secondary sexual characteristics, the presence of a low testicular volume (<4 mL) indicates the presence of an abnormal pubertal process. However, it is difficult to distinguish the presence of hypogonadotropic hypogonadism or a constitutional delay in puberty based on the testicular volume alone, and gonadotropin levels are not much help. Measurements of anti-Müllerian hormone (AMH) and inhibin B, markers of Sertoli cell function, could help guide the diagnosis. Indeed, the presence of age-related normal levels of AMH and inhibin B suggests constitutional delay, while subnormal levels suggest congenital hypogonadotropic hypogonadism. However, to date, the diagnosis of certainty can only be made after puberty or by genetic analysis [7].

In addition to assessing testicular volume, scrotal palpation allows for the detection and suspicion of many other abnormalities. For example, the presence of acute scrotal pain associated with the sensation of scrotal mass or swelling may lead to suspicion of testicular torsion or epididymitis-orchitis. However, in these cases, only a scrotal ultrasound scan allows a differential diagnosis of certainty.

Another cause of acute testicular pain is torsion of the testicular appendage. In these cases, the pain is less acute and the scrotal edema less pronounced than in testicular torsion, and the sign of the blue dot, indicative of the necrosis of the testicular appendix, may sometimes be evident [8].

In addition to the palpable masses associated with pain, testicular palpation may allow us to suspect, especially in young adults with a history of cryptorchidism or other risk factors, the presence of tumor masses that generally present themselves as single and firm nodularity [8]. 

Among the scrotal palpable masses, the hydrocele, which can be differentiated from other testicular masses by transillumination of the fluid with a torch, and the varicocele are also recognizable [8]. The latter is the dilation of the pampiniform plexus veins and in advanced forms, is related to abnormalities of seminal fluid and infertility. In particular, varicocele must be evaluated in orthostatism and clinically it can be classified according to the Dubin-Amelar classification in three different degrees. In detail, varicocele is first degree when it is palpable only at the Valsalva’s maneuver, second degree when it is palpable even at rest, and third degree when visible through the scrotal skin [9]. 

The scrotal physical examination allows for the evaluation of the integrity of the epididymal structures and highlights the possible presence of epididymal cysts or vas deferens agenesis which could cause obstructive azoospermia [10]. Similarly, digital-rectal exploration of the prostate could highlight the presence of median prostate cysts, in turn, considered a potentially reversible cause of infertility due to obstruction. In these cases, semen analysis with evidence of low seminal fluid pH and low volume further supports the diagnostic hypothesis [10]. 

Finally, exploration of the inguinal canal during scrotal palpation may also allow for evidence of an inguinal hernia [8]. 

### 2.3. Palpatory Signs in Other Endocrinologic-Andrological Diseases

In evaluating the child’s development, a key role is also played by the evaluation of the size of the penis. In particular, the presence of a penis diameter <2.5 standard deviation (SD) below the average for age is indicative of micropenis. For a correct measurement, the penis should be measured when it is fully extended, with the foreskin retracted and holding the glans between the thumb and forefinger. The measurement is taken from the pubic branch to the distal tip of the glans penis on the dorsal side. Crucial during the measurement is to press the suprapubic fat pad as much as possible inward to exclude a “buried penis”. Since penile development largely depends on adequate development of the hypothalamic-pituitary-gonadal axis, any conditions of hyper- or hypogonadotropic hypogonadism must be excluded in cases of micropenis. Furthermore, growth hormone deficiency/resistance or abnormalities of the androgen receptor function may also be associated with this condition. When all diseases are excluded, one can speak of idiopathic micropenis [11].

Finally, a mention deserves gynecomastia. It is the benign proliferation of mammary glandular tissue in men. Although it is common and generally unrelated to abnormalities in childhood and adolescence, it can be a sign of many diseases in adulthood. Gynecomastia occurs when there is an imbalance between testosterone (T) and 17ß-estradiol (E_2_) [12,13]. Therefore, all forms of central or primary hypotestosteronemia can lower the T/E_2_ ratio and hence must be excluded. Other common causes include obesity, which increases estrogen levels due to the aromatase action of visceral fat. In turn, estrogens increase SHBG levels and exert negative feedback at the hypothalamic-pituitary level with a consequent decrease in testosterone levels and a further reduction in the T/E_2_ ratio. Other causes of gynecomastia are renal failure, which causes hypotestosteronemia and hyperprolactinemia, and liver failure, which is associated with increased SHBG levels and consequent reduction in testosterone levels and impaired estrogen metabolism. Furthermore, the use of anti-androgenic drugs, such as spironolactone, could play a role in these patients. Another serious condition, which can be suspected in presence of gynecomastia, is a testicular tumor secreting hCG. This hormone, having an LH-like action, stimulates the activity of testicular aromatase, increasing estrogen secretion that results in breast proliferation. Therefore, in the case of gynecomastia, a careful evaluation of the gonadal hormone profile (LH, FSH, total testosterone, SHBG, albumin, and E_2_), the hepatic and renal function, and a testicular ultrasound should be carried out for an early diagnosis of these diseases.

Furthermore, in the case of gynecomastia, a careful drug history is necessary to exclude the use of drugs that can alter the T/E_2_ ratio, such as antiandrogens, and psychoactive drugs (haloperidol, etc.), protonic pump inhibitors, etc. Furthermore, the abuse of illicit substances, including cannabis and alcohol, which have a direct inhibitory effect on the hypothalamus-pituitary-testicular axis, must also be excluded [12,13]. Finally, an abnormal thyroid function can also cause gynecomastia. Thyrotoxicosis by increasing SHBG levels leads to a reduction in free testosterone levels and a compensatory increase in testicular steroidogenesis and aromatase activity, resulting in a T/E_2_ imbalance. On the other hand, hypothyroidism lowers testosterone levels by causing hyperprolactinemia [12,13]. The main goal of the breast examination is to distinguish true gynecomastia from pseudo gynecomastia (or lipomastia), which is extremely common in obese individuals. It is also necessary to rule out possible breast cancer. It is necessary to inspect the skin and nipple and to evaluate, the size, consistency, and laterality of the lesion, as well as the presence of any adenopathy in the axillary cavity. Finally, the presence of galactorrhea must also be sought by lightly pressing the gland. The latter manifestation is rare in patients with gynecomastia, but when present it may indicate the presence of hyperprolactinemia. Glandular tissue is often bilateral and is perceived as an elastic, soft, sometimes tender mass, usually located concentrically behind the areola. In contrast, breast cancer is typically a unilateral hard mass located primarily outside the areolar area, occasionally accompanied by skin changes, ulceration, nipple retraction or bleeding, and possible axillary lymphadenopathy. The differential diagnosis between gynecomastia and lipomastia becomes more complex in the case of gynecomastia, which has been present for several years and has undergone a process of fibrosis. In this case, the instrumental diagnostic examination allows a differential diagnosis between the two conditions [13].

## 3. Main Dermatological Signs Associated with Endocrine Diseases

### 3.1. Dermatologic Signs in Cushing Syndrome

Several dermatological signs contribute to orienting the suspicion towards an endocrine disease. For instance, striae rubrae and the buffalo hump are among the cutaneous signs with the greatest predictive diagnostic value. Both are strongly related to Cushing’s syndrome. The buffalo hump is caused by the relevant role of hypercortisolism in the reorganization of body fat, which leads to centripetal obesity and the accumulation of fat on the face (resulting in the typical moon face), in the supraclavicular region, and the retronucal region (resulting in the buffalo hump). Striae rubrae are purple streaks characterized by a width >1 cm, commonly localized on the abdomen and lower flanks. They can also occur on the upper arms, shoulders, armpits, breasts, hips, buttocks, and upper thighs [14]. Their pathogenesis derives from the ability of excess glucocorticoids to inhibit the function of fibroblasts involved in the production of extracellular matrix proteins (such as proteoglycans) that play a fundamental for the skin. Furthermore, collagen turnover is impaired. As a result, the skin becomes thinner, atrophic, brittle, and less elastic, favoring the formation of stretch marks [15]. These striae are virtually pathognomonic of Cushing’s syndrome since they differ from those of obese or pregnant women which are generally pink or reddish and thinner [14]. If the presence of Cushing’s syndrome is suspected, other skin manifestations may also allow differentiation between ACTH-dependent and independent forms. In fact, skin hyperpigmentation and hirsutism are often present in the former, although hirsutism can also be present in adrenal adenomas or carcinomas due to the mixed production of cortisol and androgens [14]. When the evolution and severity of symptoms is particularly rapid, with a predominance of catabolic symptoms with purple striae, pressure sores, osteoporosis, profound hypokalemia, and severe hypertension with edema, an ectopic Cushing syndrome should be searched for [16].

### 3.2. The Role of Skin Pigmentation Changes in Endocrine Diseases

Changes in skin pigmentation play an important role in endocrine diseases. Several and even rare syndromes are associated with skin manifestations and endocrine dysfunctions. For example, café-au-lait macules are often found in numerous rare genetic syndromes that are associated with endocrinological abnormalities. They are often the first clinical manifestation of numerous RASopathies such as neurofibromatosis type 1 and Noonan syndrome [17]. Also, in McCune Albright syndrome the first clinical manifestation is often represented by these spots, which are usually fewer in number and larger in diameter, with darker pigmentation and more irregular borders, than those in patients with NF. They are mainly located in the posterior neck, base of the spine, trunk, and face [18]. Other skin pigmentation, such as lentiginosis, may also be present in forms of Noonan syndrome associated with lentiginosis (formerly known as LEOPARD syndrome) [17]. Even more rarely, the presence of freckles and hyperpigmented skin macules (e.g., ephelides or blue nevi) in association with the presence of myxomas (cutaneous or otherwise) and nodular hyperplasia of the adrenal gland or GH- or ACTH-secreting pituitary adenomas suggests the presence of a Carney complex [19]. 

Other skin manifestations are much more common. For example, skin hyperpigmentation, mentioned above for ACTH-dependent Cushing disease, is also a sign of primary adrenal insufficiency. In this case, elevated levels of ACTH can cross-react with the melanocortin receptor 1 on melanocytes, stimulating them. Furthermore, since ACTH originates from proopiomelanocortin (POMC), an increase in melano-stimulatory hormones results from the cleavage of POMC. In detail, hyperpigmentation is usually generalized. However, it is more pronounced in the areas most exposed to the sun. Moreover, surface of the skin under pressure (elbows and knees), around the nipples, and the genital region, are also affected. It can also be present on mucosal surfaces such as the tongue, oral mucosa, and the inner part of the lips [20]. Another disease with pronounced skin hyperpigmentation is Nelson’s syndrome, which results from elevated ACTH levels that generally occur in patients with treatment-refractory Cushing’s disease undergoing bilateral adrenalectomy [21].

Skin hypopigmentation and, in particular the presence of vitiligo, should also be an alarm sign for the patient in whom the endocrinologic disease is suspected. Vitiligo is closely associated with the development of autoimmune thyroiditis, so much so that in patients with this skin sign, the search for anti-thyroid antibodies and the evaluation of the function of this gland is recommended [22]. However, other autoimmune diseases such as diabetes mellitus type 1 and Addison’s disease can be associated with vitiligo. When several of these endocrine autoimmune manifestations are concomitantly present, an autoimmune polyendocrine syndrome can be diagnosed. Therefore, vitiligo could be a sign of a much more complex clinical condition [23].

### 3.3. Piliferous Evaluation in Endocrinology

Another sign that is a frequent reason for endocrinological consultation, affecting between 4.3 and 10.8% of women, is hirsutism. It should be assessed using the Ferriman-Gallwey index (FGi). An FGi > 8 is indicative of hirsutism [24]. However, race and ethnicity must also be considered in the evaluation of FGi. Scores of 8 or higher are considered hirsutism in white and black British and U.S. women. Conversely, in Mediterranean, Hispanic, and Middle Eastern women, a score of 9 or higher is considered abnormal. Scores of six or higher are indicative of hirsutism in South American women. Finally, in Asian women, even scores equal to or higher than 2 also allow the diagnosis of hirsutism to be made. Scores up to 15 indicate mild hirsutism, while scores above 25 indicate severe hirsutism. The main limitation of this system is that its evaluation is subjective. It also does not take into account any locally high scores or reductions in scores resulting from previous cosmetic treatments [25]. Several conditions associated with increased production of androgens can be responsible for hirsutism. In detail, the presence of this sign in association with ovulatory dysfunction and/or polycystic ovarian morphology should lead us to suspect PCOS and, in this case, it will be necessary to evaluate the metabolic features and the cardiovascular risk of these patients [24]. In fact, in these women, insulin resistance and hyperinsulinemia by acting directly on the ovarian theca cells may contribute to the pathogenesis of hirsutism [26]. Another disease that can cause hirsutism is congenital adrenal hyperplasia, non-classical variant. To confirm its presence, it is necessary to measure 17αOH-progesterone levels, which must exceed 10 ng/mL at baseline or after stimulation with ACTH. Once the disease has been confirmed biochemically, it is useful to search for 21-hydroxylase mutations. Rarer are the forms associated with 11ß-hydroxylase mutations, which are characterized by high levels of 11-deoxycortisol [27]. Once these conditions are ruled out, we may be faced with idiopathic hyperandrogenism if androgen levels are high or idiopathic hirsutism if these hormones are in the normal range. It is important to exclude the intake of drugs that cause hirsutism, such as phenothiazines, glucocorticoids, and anabolic agents. Finally, before diagnosing hirsutism as idiopathic, it is also important to rule out other rare endocrine diseases associated to hirsutism. These include the aforementioned Cushing’s syndrome, acromegaly, or androgen-secreting adrenal or ovarian tumors [24]. However, in the case of androgen-secreting malignant tumors, we are often faced with forms of severe hirsutism that do not respond to therapy and which are often accompanied by other signs of virilization such as hypertrophy of the clitoris, deepening of the voice, and increased trophism of the muscle masses [25].

Hirsutism must be differentiated from hypertrichosis. While in hirsutism, there is an overgrowth of androgen-sensitive hair in a male pattern, in hypertrichosis there is a growth of terminal hair in areas where they are not normally present, regardless of whether they are regions sensitive to the effects of androgens. Therefore, in the management of a patient who comes to the clinician’s attention, it is first necessary to distinguish between these two conditions. There are several congenital and acquired causes of hypertrichosis and among them are also recognized endocrine causes. In particular, hypothyroidism can be associated mainly in children with an alteration of the hair on the scalp, which can become coarse, dull, and brittle. This condition reverses with the initiation of L-thyroxine replacement therapy. In addition, areas of hypertrichosis can be observed in hyperthyroidism at the plaques of pretibial myxedema. Another condition that can lead to hypertrichosis and is also associated with endocrine abnormalities is represented by the POEMS syndrome (polyneuropathy, organomegaly, endocrinopathy, M protein, and skin changes) [28].

Unlike hirsutism, in men, the presence of a reduction in the percentage of body hair, especially in the pubic and axillary region, as well as a reduction in the growth rate of the beard and therefore in the frequency of shaving, could be signs of low testosterone levels. Therefore, in these cases, the clinician must investigate the presence of decreased libido, erectile dysfunction, and other signs and symptoms compatible with hypogonadism [29].

### 3.4. Acne and Its Relevance in Endocrine Diseases

Closely associated with hirsutism from the etiopathological point of view is acne. Indeed, this condition too can be seen in diseases that increase androgen production such as PCOS, NCCAH, and androgen-secreting tumors. Furthermore, in cases of hypercortisolism such as in Cushing’s syndrome, steroid acne can be observed. It is characterized by erythematous, monomorphic papules or small pustules distributed along the upper part of the trunk, the proximal upper extremities, the neck, and the face [30].

### 3.5. Acanthosis Nigricans in Endocrinology

Of extreme interest among the dermatological signs of endocrine disorders is acanthosis nigricans, described in Section 5 for the close association between the pathogenesis of this sign and metabolic alterations [31].

### 3.6. Other Signs Associated with Rare Endocrine Diseases

Among the dermatological manifestations that best correlate with an endocrinological disease is the necrolytic migratory erythema (NME). Indeed, this is often the first sign that leads to suspect the presence of a glucagonoma, a rare endocrinologic disorder. It appears as a bullous and itchy dermatosis that evolves over a few weeks into patches or plaques with irregular edges, crusts ulcerations, and peeling. When these lesions fade, the skin sometimes takes on an eczematous and psoriasiform appearance. It may be diffuse or isolated to the perioral region, trunk, groin, intergluteal region, genital area, and lower extremities. The pathogenesis may in part be attributed to hyperglucagonemia, which results in increasing hepatocyte gluconeogenesis and lipolysis leading to hypoaminoacidemia, which in turn is associated with NME. Furthermore, hyperglucagonemia can contribute to increasing levels of arachidonic acid, prostaglandins, and leukotrienes, predisposing the inflammatory reaction typical of this dermatosis. Confirming the role of hyperglucagonemia, surgical removal of glucagonomas, or stabilization of glucagon levels with drugs helps resolve the rash. However, other mechanisms could also contribute to its pathogenesis and explain why even in pseudoglucagonoma syndrome, where glucagon levels are normal, NME may still be present. In particular, malnutrition can contribute to the development of NME. The deficiency of zinc, protein, amino acids, and essential fatty acids can cause NME-like dermatitis [32].

Finally, it should be mentioned the skin flushing, in particular on the face, telangiectasia, and pellagra-like lesions that, in a patient with profuse diarrhea and asthma-like symptoms, can orient the clinician toward the suspicion of a carcinoid syndrome [33].

## 4. Main Anthropometric Signs Associated with Endocrine Disorders

### 4.1. The Enuchoid Habitus in the Evaluation of the Hypothalamic-Pituitary-Gonadal Axis

Among the anthropometric signs that best correlate with the presence of endocrine diseases is the eunuchoid habitus. Tall stature, underweight, long upper and lower limbs, and an arm span of more than 5 cm longer than height characterize this phenotype. It can also be associated with changes in primary and secondary sexual characteristics and/or metabolic parameters and altered fat mass distribution [34]. Regarding the latter, the distribution of fat largely depends on the levels of circulating sex hormones. In particular, testosterone hinders adipogenesis while estrogens stimulate the proliferation of preadipocytes both in the subcutaneous and visceral abdominal level, and progestins stimulate their differentiation. Therefore, testosterone is essential not only for the trophism of muscle masses but also for the different distribution of fat between men and women. When hypogonadism occurs at developmental age, testosterone deficiency is associated with relative hyperestrogenism that promotes fat deposition at the lower body level, particularly around the hips and thighs, leading to gynoid obesity [35]. The presence of eunuchoid habitus, associated or not with gynoid obesity, allows the endocrinologist to suspect the diagnosis of hypogonadism, which can then be confirmed by investigating the sex steroid profile of these people. Furthermore, the measurement of the gonadotropin levels reveals whether the hypogonadism is of testicular or hypothalamic-pituitary origin [34].

### 4.2. Main Anthropometric Signs in Diseases of the GH-IGF1 Axis

The presence of an increase in the size of the extremities, associated with pronounced protrusion of the frontal bumps, arching, and thickening of the eyebrows, enlargement of the nose and ears, thickening of the lips, skin wrinkles, nasolabial folds, and mandibular prognathism lead to dental malocclusion and increased interdental spacing, an acromegaly may be suspected. These alterations are partly attributable to soft tissue swelling which in turn is associated with the deposition of glycosaminoglycans, increased connective tissue collagen production, and the edema that occurs in this condition. Macroglossia is also common and contributes to the development of obstructive sleep apnea, which is a major complication of this disease. All of these signs are often subtle in their appearance and therefore general practitioner, as well as patients and their families, do not pay attention to these changes [36]. 

When GH overproduction occurs before epiphyseal cartilage welding, a condition of gigantism is determined, characterized by a height greater than three SD per age or more than two SD beyond the target height calculated from the parental height. Furthermore, also in these patients, there is acral enlargement and alteration of facial features [36]. 

In case of a reduction in height from four to ten standard deviations from the average height for age a GH deficiency or a resistance to its action like GH insensitivity syndromes (such as Laron syndrome) could be suspected. In detail, this reduction seems to be more marked in patients with GH insensitivity syndromes than in those with congenital GH deficiency. In addition, patients with Laron syndrome show an upper to lower segment ratio above normal for sex and age, denoting short limbs for the trunk size. Other signs that allow suspecting GH insensitivity syndromes, as well as GH deficiency, are some common facial abnormalities (e.g., protruding forehead due to reduced development of the face bone, sparse and bristly hair, and crowded teeth that frequently become decayed) [37]. These signs are often associated with a reduction in volume of genitalia with delayed puberty, although the complete pubertal development would seem to have been achieved regularly in these patients [38].

### 4.3. Anthropometric Signs Associated with Hypothyroidism

In patients with periorbital edema and loss of the outer third of the eyebrows, a lowering of the upper eyelid, nose enlargement, lips thickening and macroglossia other signs and symptoms of hypothyroidism should be looked for. These signs are mostly attributable to the accumulation of mucopolysaccharides in the dermis, which cause the so-called myxedema or secondary to a decrease in sympathetic stimulation (such as lowering of eyelid) [4]. 

## 5. Main Physical Sings Associated with Metabolic Disorders

### 5.1. Waist Circumference and Waist-to-Hip Ratio as Predictors of Cardiovascular Risk

The waist circumference is among the signs with an important role from the prognostic point of view. It is a useful tool for assessing the severity of obesity. Indeed, differently from the body mass index (BMI), waist circumference allows the identification of patients with an increased risk of developing obesity-related chronic complications, such as cardiovascular diseases and diabetes mellitus [39]. Indeed, this parameter correlates better with visceral adipose tissue, which in turn has a close relationship with cardiovascular and metabolic risk. Therefore, a measurement of the waist circumference must always be carried out in the semiological evaluation of a patient as it represents a sign of pivotal relevance [39].

Similarly, increased waist-to-hip ratio (WHR) correlates significantly with cardiometabolic risk and the likelihood of developing myocardial infarction. It shows the presence of visceral adiposity even in those subjects where there is no obvious increase in body weight [40]. In particular, as established by the World Health Organization, the cardiometabolic risk is substantially increased when the waist circumference is >102 cm in men and >88 cm in women. As regards WHR, the risk is significantly increased when it is >0.9 in men and >0.85 in women [41].

Moreover, in women of childbearing age, an increase in these two indices would also appear to correlate with an increased risk of having polycystic ovary syndrome (PCOS), thus directing the clinician to study this condition [42].

### 5.2. Acanthosis Nigricans as a Sign of Metabolic Dysfunction in Endocrine Diseases

Acanthosis nigricans is another metabolic sign that correlates significantly with obesity and insulin resistance, allowing suspecting disorders related to this sign, such as diabetes mellitus, metabolic syndrome, and PCOS [31]. Its prevalence varies significantly among populations being much more frequent in African-Americans followed by Hispanics, Asians, and much fewer Caucasians [43]. Acanthosis nigricans is characterized by the presence of dark, velvety papillomatous plaques of hyperkeratosis. The pathogenesis depends on hyperinsulinemia, which directly and indirectly stimulates the IGF1 receptor (IGF1R) on the surface of keratinocytes and fibroblasts, stimulating their proliferation [31]. Adequate pharmacological anamnesis might also be useful in the clinical characterization of this sign, since the use of drugs that cause hyperinsulinemia, such as glucocorticoids, niacin, estrogen-progestogen therapies, and protease inhibitors, may lead to its appearance [44]. In addition to metabolic syndrome and obesity, acanthosis nigricans may be present in some other endocrine diseases that impair glucose metabolism. These include Cushing syndrome and acromegaly. Male hypogonadism, which causes visceral adiposity and consequently metabolic dysfunctions, can also be associated with its presence. Therefore, the presence of acanthosis nigricans in the neck, eyelids, lips, axillae, mucosal surfaces, dorsal hands, and flexural areas in the groin, knees, and elbows should always be sought in patients with suspected endocrine/metabolic disorders [43].

### 5.3. Signs Associated with Lipid Metabolism Abnormalities 

As regards lipid metabolism, various signs direct the clinician towards the presence of dyslipidemia and in some cases towards the suspicion of a hereditary condition. These include xanthomas, which are lesions located in the connective tissue of the skin or tendons and fascia. They are made of macrophages that incorporate LDL cholesterol particles, leading to the formation of foam cells [45]. The presence of some types of xanthomas sometimes plays a pathognomonic role, as in the case of dysbetalipoproteinemia, characterized by the presence of striatum palmar xanthoma [45]. Tendon and tuberous xanthomas are characteristic of autosomal dominant hypercholesterolemia, especially if they appear at a young age [46]. However, they also occur in some rare conditions, such as cerebrotendinous xanthomatosis and familial β-sitosterolemia. Furthermore, the presence of tendon xanthomas in familial hypercholesterolemia appears to be associated with a two to four times greater risk of cardiovascular disease [45]. On the other hand, eruptive xanthomas are frequently found in severe hypertriglyceridemia and carry a high risk of acute pancreatitis or type 2 diabetes mellitus [45].

The most frequent form of xanthoma is eyelid xanthelasma. When present in children in association with corneal arch and tuberous or tendon xanthomas, autosomal dominant hypercholesterolemia maybe suspected. Generally, however, their frequency increases in the population over 50 years of age and their presence has a negative predictive role since they are associated with the presence of significantly higher levels of atherogenic LDL and a significantly higher risk of atherosclerosis than controls. Therefore, patients with these formations must be carefully monitored for the prevention of cardiovascular risk [47].

Another sign related to hyperlipidemia is the corneal arch. This is caused by the deposition of lipids in the peripheral region of the corneal stroma. The width of the ring appears to be related to the severity of dyslipidemia and the duration of the condition. This deposition results in the formation of a gray-white or yellowish ring approximately 1 mm in diameter separated from the limbic margin by the Vogt lucid interval, a 0.3–1 mm wide area of the clear cornea. The deposition of lipids in the periphery of the cornea is due to the fact that this area is the one that receives most of the perfusion of the limbal vascular system. It is generally bilateral, whereas unilateral forms are seen on the contralateral side to that of a carotid artery occlusion, reinforcing the importance of limbic vascularity in the genesis of this condition. Prevalence increases with age and is higher in men. In addition to age, its presence is also related to alcohol intake, diabetes mellitus, smoking, blood pressure, BMI, and obesity. The correlation with dyslipidemia is demonstrated by the fact that ring extension appears to correlate with the duration and severity of LDL cholesterol levels and with an altered LDL/HDL cholesterol ratio [48]. As with xanthelasma and tendon xanthomas, its onset at a young age suggests the presence of familial hypercholesterolemia, which should be investigated by genetic testing and promptly treated [48].

Again, from the metabolic point of view, another sign with a prognostic and predictive role for severer diseases is hepatomegaly. It is often related to the accumulation of fat in the liver. When present, therefore, it is necessary to exclude the possibility of alcohol abuse and, if not, the presence of non-alcoholic fatty liver disease (NAFLD) can be suspected. This condition is strongly associated with obesity, metabolic syndrome, and an increased risk of developing serious diseases, such as liver cirrhosis and hepatocarcinoma. Furthermore, patients with NAFLD appear to have a higher prevalence of DM, which in turn is associated with worsening of the clinical condition. Diagnosis is generally based on ultrasound examination and evaluation of non-invasive scores, although the diagnosis of certainty can only be made by biopsy. Therefore, in obese patients, the presence of hepatomegaly is associated with an increased risk of metabolic and hepatic comorbidities [49]. Another rare endocrine disease associated with visceromegaly, including hepatomegaly, is acromegaly. However, other signs and symptoms generally allow one to suspect its presence (see Section 4) [50].

## 6. Other Signs Associated with Endocrine Diseases

### 6.1. Signs of Hypocalcemia

Trousseau and Chvostek signs are two fundamental signs to be sought in case of suspected hypocalcemia. Indeed, they are caused by the increased neuromuscular irritability, in turn related with the low calcium levels. When hypocalcemia-induced neuromuscular hyperexcitability is latent, looking for these two signs allows the diagnosis to be confirmed. The first can be evoked with a sphygmomanometer, which is inflated slightly above the patient’s systolic pressure. Ischemia resulting from occlusion of the brachial artery results in flexion of the wrist joint and metacarpophalangeal joints, flexion of the fingers, and adduction of the thumb (obstetrician’s hand). In the sign of Chvostek, on the other hand, the percussion of a point located in front of the earlobe and under the zygomatic process causes contraction of the muscles of the ipsilateral face [51].

### 6.2. Ophtalmological Evaluation in Invasive Diseases of the Hypothalamic-Pituitary Unit

In patients who report reduced visual acuity up to bitemporal hemianopsia, compression of the chiasma by a pituitary adenoma may be suspected. Indeed, large adenomas can compress the optic chiasm and consequently the retinal nasal fibers, causing a reduction in the visual field. This reduction is often associated with diplopia. The presence of this additional sign indicates the extension of the adenoma towards the cavernous sinus and the consequent compression of the oculomotor nerves. Therefore, all of these signs also have a prognostic role regarding the radicality of a possible surgical removal, since they indicate the extension of adenomas towards important cerebral structures [52]. 

## 7. Concluding Remarks

In conclusion, physical examinations still play a key role in good endocrine clinical practice today. In fact, since hormones are essential in regulating the physiological processes of various organs and apparatuses of our body, their imbalance can lead to the onset of various physical signs that can guide the clinician towards specific diagnoses. This reduces the time and cost required to reach a diagnosis. Additionally, in some cases, they may be among the first physical signs of serious illnesses or play an important prognostic role in the development of future morbidity and mortality (Table 1).

## Figures and Tables

**Table 1 jcm-11-02598-t001:** Main physical signs in the clinical evaluation of patients with endocrine/metabolic diseases and their possible diagnostic significance.

Category	Type of Sign	Associated Findings	Diagnostic Suspicion
Palpatory signs	Goiter	Hormonal dysfunction	Hypothyroidism or Hyperthyroidism
	Painful goiter	Fever, initial symptoms of thyrotoxicosis, and previous viral infection	Subacute thyroiditis
	Firm nodule	Lymphadenopathy	Thyroid cancer
	Pemberton’s sign		Endothoracic goiter
	Mobile midline mass of the neck	Movement with protrusion of the tongue	Thyroglossal duct cysts
	Pretibial myxedema	Thyrotoxicosis	Hyperthyroidism
	Small testis	Reduced activation of the hypothalamic-pituitary-gonadal axis or reduced function of testis	Pubertal growth retardation or hypogonadism
	Small firm testis	Hypergonadotropic hypogonadism	Prepubertal primary testiculopathy (such as Klinefelter syndrome)
	Acute scrotal pain	Scrotal swelling	Testicular torsion, orchitis, epididymites
		Blue dot sign	Torsion of the testicle appendix
	Testis firm nodule	History of cryptorchidism and/or young age	Testis cancer
	Scrotal mass	Liquid transillumination with a torch	Hydrocele
		Reflux at Valsalva’s maneuver or visible varices	Varicocele
	Absence of vas deferens at palpation		Obstructive azoospermia and Cystic Fibrosis
	Micropenis	Low gonadotropin and testosterone levels	Hypogonadotropic hypogonadism
		High gonadotropin and low testosterone levels	Hypergonadotropic hypogonadism
		Low GH and IGF1 levels	GH deficiency
		High GH and low IGF1 levels	GHIS
	Gynecomastia	Low gonadotropin and testosterone levels	Hypogonadotropic hypogonadism
		High gonadotropin and low testosterone levels	Hypergonadotropic hypogonadism
		High prolactin levels and rarely galactorrhea	Prolactinoma
		Altered kidney parameters with high prolactin and low testosterone levels	Renal failure
		Altered liver parameters with low testosterone and high estrogen levels	Liver failure
		Testis firm nodule and βhCG high levels	Testis cancer
			Exclude use of drugs altering testosterone levels and illicit drugs abuse
Dermatological signs	Striae rubrae	Buffalo hump, moon face, high urinary cortisol levels, and lack of suppression in Nugent’s test	Cushing’s syndrome
		Predominance of catabolic symptoms with pressure sores, osteoporosis, profound hypokalemia, and severe hypertension with edema, high urinary cortisol levels and lack of suppression in Nugent’s test	Ectopic Cushing’s syndrome
	Hyperpigmentation	Fatigue, dizziness, nausea, vomiting, low blood pressure, high ACTH levels	Primary adrenal insufficiency
		History of bilateral adrenectomy for refractory Cushing’s disease	Nelson’s syndrome
	Vitiligo		Check for thyroid autoimmunity and for symptoms associated to other gland autoimmune disease
	Hirsutism	Ovulatory dysfunction and/or polycystic ovarian morphology	PCOS
		17αOH-progesterone levels > 10 ng/mL at baseline or after stimulation with ACTH	NCCAH
		Only hyperandrogenism	Idiopathic hyperandrogenism
		No hyperandrogenism or other signs	Idiopathic hirsutism
		Signs of virilization such as hypertrophy of the clitoris, deepening of the voice, and increased trophism of the muscle masses	Androgen-secreting tumors
	Hypertrichosis	Other signs of hypothyroidism	Hypothyroidism
		Pretibial myxedema	Hyperthyroidism
		Polyneuropathy, organomegaly, endocrinopathy, M protein, and other skin changes	POEMS syndrome
	Acne	Ovulatory dysfunction and/or polycystic ovarian morphology, hirsutism	PCOS
		17αOH-progesterone levels > 10 ng/mL at baseline or after stimulation with ACTH, hirsutism	NCCAH
		Buffalo hump, moon face, striae rubrae, high urinary cortisol levels and lack of suppression in Nugent’s test	Cushing’s syndrome
	NME	High glucagon levels, hypoaminoacidemia	Glucagonoma
	Skin flushing	Telangiectasia, and pellagra-like lesions and asthma like symptoms	Carcinoid syndrome
Anthropometric signs	Eunuchoid habitus	Low gonadotropin and testosterone levels, gynoid obesity	Hypogonadotropic hypogonadism
		High gonadotropin and low testosterone levels, gynoid obesity	Hypergonadotropic hypogonadism
	overgrowth of the extremities	Pronounced protrusion of the frontal bumps, arching and thickening of the eyebrows, enlargement of the nose and ears, thickening of the lips, skin wrinkles, nasolabial folds, and mandibular prognathism, macroglossia, visceromegaly	Acromegalia
	Short stature, protruding forehead, sparse and bristly hair, and crowded teeth	High GH and low IGF1	GHIS
		Low GH and IGF1	GH deficiency
	Periorbital edema	Loss of the outer third of the eyebrows, lowering of the upper eyelid, enlarged nose, thickened lips, macroglossia	Hypothyroidism
Sings associated with metabolic disorders	Waist circumference and WHR	Obesity	Increased cardiovascular risk
		Ovulatory dysfunction and/or polycystic ovarian morphology, hirsutism	PCOS
		Decreased libido, erectile dysfunction, low testosterone levels	Hypogonadism
	Acanthosis nigricans	Hyperglycemia	Diabetes
		Ovulatory dysfunction and/or polycystic ovarian morphology, hirsutism	PCOS
		Buffalo hump, moon face, striae rubrae, high urinary cortisol levels, and lack of suppression in Nugent’s test	Cushing’s syndrome
		Decreased libido, erectile dysfunction, low testosterone levels	Hypogonadism
		Overgrowth of the extremities, pronounced protrusion of the frontal bumps, arching and thickening of the eyebrows, enlargement of the nose and ears, thickening of the lips, skin wrinkles, nasolabial folds, and mandibular prognathism, macroglossia, visceromegaly	Acromegalia
	Palmar xanthoma		Dysbetalipoproteinemia
	Tendon and tuberous xanthomas	Young age, high cholesterol levels	Autosomal dominant hypercholesterolemia
	Eruptive xanthomas		Severe hypertriglyceridemia
	Corneal arch	Young age, high cholesterol levels	Autosomal dominant hypercholesterolemia
Other signs	Trousseau sign	Paresthesias	Hypocalcemia
	Chvostek sign	Paresthesias	Hypocalcemia
	Visual reduction up to bitemporal hemianopsia	Diplopia	Optic chiasm compression by pituitary adenoma

## Data Availability

Data sharing not applicable.

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
