# Peer review of "Physical Examination for Endocrine Diseases: Does It Still Play a Role?"

_jcm, 2022, doi:10.3390/jcm11092598_

Round 1
Reviewer 1 Report
It is refreshing to read a paper about physical examination as the basis of our diagnostic, instead of focus on new diagnostic tests and techniques.
To improve readability of the manuscript I advise to improve priority/focus, e.g. by using more highlights or subcatagories. Most endocrinologists are not the prime physisian for diagnosing and treating metabolic syndrome related disorders. Hence, I would suggest to start the manuscript with endocrine/hormone related topics instead of the metabolic abnormalities.
The manuscript also needs extensive editing of English language and style. Many sentences and paragraphs are difficult to read due to incorrect English language and grammar.
Line 39: 'turning to the endocrinologist for treatment of obesity' : most of the times it is not the endocrinologist patients are refered to, but internal medicine specialists or obesity specialists. In fact, waist circumference should not only be measured in patients referred for obesity, but in every patient in the outpatient clinicic with overweight/obesity. For interpretation of the waist circumference and WHR, can you present reference values?
Acanthosis nigricans: many aspects/etiologies are described, omitting the fact that etnicity also plays a role. In addition, the relation between drugs and acanthosis is not very strong, since most data are based on (single) case reports. This should be stated differently in the manuscript. Acanthosis should also be ckecked in the axillae (add this to line 60).
The paragraph on dyslipidaemia is very difficult to read, please rewrite. It would also be helpful to descripe the corneal arch in more detail: how does it appear? Can you add an image?
Line 100-103: The description of acromegaly is quite out of context here. Since acromegaly is also characterized by clinical featurs visible on examinations, it would be useful to describe these features as well, instead of referring to the literature. You can also present this condition in more detail in paragraph 3.
Paragraph 4: Acanthosis nigricans is not mentioned in this paragraph, despite being a major dermatological sign. You can mention this breefly and refer to paragraph 1.
Paragraph 5, line 186-189: the usefulness of AMH and inhibin B in discriminating hypogonadotropic hypogonadism from constitutional delay is not very clear yet. In clinical practice, the final diagnosis can only be made after puberty (with of without induction with testosterone or estrogens). In addition AMH values can be quite variable in pediatric patients even within one patient.
Gynecomastia: line 213, use of other substances such as cannabis are also related to the development of gynecomastia. Many adolescents and young adults use cannabis nowadays.
Figure 1: I admire your attempt to include all signs in one figure. However, figure one is a very complex and difficult to read figure. For Acanthosis only one relation is presented in the figure, while for other sings more etiologies are presented. It would be helpful to present the abbreviations in the legenda of the figure.
In the figure testicular volume of 1 and 2 ml are indicated, while in the text this discrimination is not made. Can you elaborate on this? Why do you make this discrimination? Why not use < 4 ml as characteristic feature, with subcatagory firm consistency?
Author Response
Answers to the comments of Reviewer 1
Manuscript ID:jcm-1633588
Comment 1: It is refreshing to read a paper about physical examination as the basis of our diagnostic, instead of focus on new diagnostic tests and techniques. To improve readability of the manuscript I advise to improve priority/focus, e.g. by using more highlights or subcatagories. Most endocrinologists are not the prime physisian for diagnosing and treating metabolic syndrome related disorders. Hence, I would suggest to start the manuscript with endocrine/hormone related topics instead of the metabolic abnormalities. The manuscript also needs extensive editing of English language and style. Many sentences and paragraphs are difficult to read due to incorrect English language and grammar.
Answer 1: We thank the reviewer for the comment. We have expanded the text of the subcategories and changed the order. The English text was extensively revised.
Comment 2:Line 39: 'turning to the endocrinologist for treatment of obesity': most of the times it is not the endocrinologist patients are refered to, but internal medicine specialists or obesity specialists. In fact, waist circumference should not only be measured in patients referred for obesity, but in every patient in the outpatient clinicic with overweight/obesity. For interpretation of the waist circumference and WHR, can you present reference values?
Answer 2: We agree with the reviewer so we deleted the sentence“contacting the endocrinologist for the treatment of obesity”. In addition, we have included the WHO established reference values for waist circumference and WHR (please, see lines 342-346).
Comment 3:Acanthosis nigricans: many aspects/etiologies are described, omitting the fact that ethnicity also plays a role. In addition, the relation between drugs and acanthosis is not very strong, since most data are based on (single) case reports. This should be stated differently in the manuscript. Acanthosis should also be checked in the axillae (add this to line 60).
Answer 3: We thank the reviewer for the comment. The section on acanthosis nigricans was revised in light of your suggestions (please,see lines 352-363)
Comment 4:The paragraph on dyslipidemia is very difficult to read, please rewrite. It would also be helpful to describe the corneal arch in more detail: how does it appear? Can you add an image?
Answer 4: This paragraph was extensively rewritten and more information on the corneal arch and a more detailed description are now given.
Comment 5:Line 100-103: The description of acromegaly is quite out of context here. Since acromegaly is also characterized by clinical features visible on examinations, it would be useful to describe these features as well, instead of referring to the literature. You can also present this condition in more detail in paragraph 3.
Answer 5: We thank the reviewer for the suggestion. A more detailed description of the physical signs of acromegaly has been added in the section “Anthropometric signs” (please,see lines 302-316)
Comment 6:Paragraph 4: Acanthosis nigricans is not mentioned in this paragraph, despite being a major dermatological sign. You can mention this briefly and refer to paragraph 1.
Answer 6:As suggested, we have also included a reference in the paragraph about the dermatological signs.
Comment 7:Paragraph 5, line 186-189: the usefulness of AMH and inhibin B in discriminating hypogonadotropic hypogonadism from constitutional delay is not very clear yet. In clinical practice, the final diagnosis can only be made after puberty (with or without induction with testosterone or estrogens). In addition, AMH values can be quite variable in pediatric patients even within one patient.
Answer7: We left the part on AMH and inhibin B because we think it represents an aspect of interest in the differential diagnosis between the two diseases. However, as suggested, we have specified in the text that to date is not possible to make a differential diagnosis of certainty based on these two data (see lines 116-117)
Comment 8: Gynecomastia: line 213, use of other substances such as cannabis are also related to the development of gynecomastia. Many adolescents and young adults use cannabis nowadays.
Answer8:As suggested, we have included cannabis as an illicit substance (see line 173).
Comment 9:Figure 1: I admire your attempt to include all signs in one figure. However, figure one is a very complex and difficult to read figure. For Acanthosis only one relation is presented in the figure, while for other signs more etiologies are presented. It would be helpful to present the abbreviations in the legend of the figure.
Answer 9: We agree with the reviewer that the figure was complex and unclear and, to a greater extent, with the addition of the parts suggested by both reviewers. Therefore, we thought to replace it with a table that, in our opinion,allows a better understanding of the clinical sign and suspected diagnoses.
Comment 10: In the figure testicular volume of 1 and 2 ml are indicated, while in the text this discrimination is not made. Can you elaborate on this? Why do you make this discrimination? Why not use < 4 ml as a characteristic feature, with subcategory firm consistency?
Answer 10: This distinction was deleted from the table
Reviewer 2 Report
Crafa et al. briefly summarize the most important features and the usefulness of the endocrine examination in clinical practice. The brief report is well written, however, in my opinion is uncompleted and some important clinical features (signs or symptoms) that endocrinologists can observed in the physical examination are missing and should be included in the review. Assuming that the authors only pretended to include the most important features of adult patients, since a list of clinical features in kids are different than adults.
Here is a list of some of the missing aspects that in my opinion should be included together with an explanation of the reasoning similar to other features already included in the manuscript. Some of them should be:
-lower limb physical examination: edema (in case of sever hypercortisolism due to ectopic Cushing, pretibial edema in sever hyperthyroidism)
-dermatosis and necrolytic migratory erythema in glucagonoma patients...
-Clinical symptoms of hypocalcemia: parestesias, Trousseau, Chovstek…,
-Additional hypogonadic signs: lower hair growth, lower shaving frequency, ginecoid habitus…, depending on the age of presentation of hypogonadism physical examination signs differed. This aspect should be elaborated more in the text.
-limitations of FGi should be included (based on race, esthetic treatments…)
-hirsutism and implications according to age of presentation, severity and acuity of presentation (clinical suspicion might differ substantially). Need to differentiate from hypertricosis.
-Acne is not mentioned, and it can be a sign of hyperandrogenism (PCOS, CAH…)
-signs of virilization in physical examination are critically important for malignant androgen tumor suspicion.
-testicular examination à look for tumours, look for varicocele and look for spermatic conduct absence… Also include micropenis and clinical suspicion
-breasts exploration à differential diagnosis between gynecomastia and lypomastia, presence of galactorrea or telorrea... might have clinical implications, and should be included
-Visual fields exploration, for a first impression of vision in a patients with pituitary adenoma with chiasma affectation.
-acromegalic features, key in the diagnostic and for the suspicion of the disease are not detailed (acral enlargement, macroglosia….also in severe hypothyroidism). Physical aspect of a gigantism and a Laron syndrome …
-Hypothyroidism: facial edema, macroglosia…
-Physical examination of thyroid cyst and thyroglossal duct cyst are not mentioned. Pemberton sign is important for the evaluation of a large endothoracic goiter
-Facial erythema, telangiectasias, might suggest carcinoid syndrome.
Minor:
Typo line 38 page 1: semiological instead of semeiological
Author Response
Answers to the comments of Reviewer 2
Manuscript ID:jcm-1633588
Comment 1:Crafa et al. briefly summarize the most important features and the usefulness of the endocrine examination in clinical practice. The brief report is well written, however, in my opinion is uncompleted and some important clinical features (signs or symptoms) that endocrinologists can observed in the physical examination are missing and should be included in the review. Assuming that the authors only pretended to include the most important features of adult patients, since a list of clinical features in kids are different than adults.
Here is a list of some of the missing aspects that in my opinion should be included together with an explanation of the reasoning similar to other features already included in the manuscript.
Answer 1: We thank the reviewer for the comments and we are sure that the article has greatly improved with her/his comments.
Comment 2:Some of them should be:lower limb physical examination: edema (in case of severehypercortisolism due to ectopic Cushing, pretibial edema in severe hyperthyroidism)
Answer 2:We have added the information on the lines 94-101 and 210-213.
Comment 3:dermatosis and necrolytic migratory erythema in glucagonoma patients.
Answer 3:As suggested, we have inserted a reference for the NME (see lines 267-281).
Comment 4:Clinical symptoms of hypocalcemia:paresthesiass, Trousseau, Chovstek
Answer 4: This wasadded in the paragraph “other physical signs” (please,see lines 411-421).
Comment 5:Additional hypogonadic signs: lower hair growth, lower shaving frequency, ginecoid habitus…, depending on the age of presentation of hypogonadism physical examination signs differed. This aspect should be elaborated more in the text.
Answer 5: As suggested by the reviewer, we have added more signs of hypogonadism in the text (see lines 259-263).
Comment 6:limitations of FGi should be included (based on race, esthetic treatments…)
Answer 6:We thank the reviewer for the comment. The text was modified by adding the FGi limits (see lines 216-223).
Comment 7:hirsutism and implications according to age of presentation, severity and acuity of presentation (clinical suspicion might differ substantially). Need to differentiate from hypertricosis.
Answer 7: As suggested, we have added information on hypertrichosis.
Comment 8:Acne is not mentioned, and it can be a sign of hyperandrogenism (PCOS, CAH…)
Answer 8: Done as suggested (please,see lines 253-257)
Comment 9:signs of virilization in physical examination are critically important for malignant androgen tumor suspicion.
Answer 9: We thank the reviewer for the suggestion. Theimportance of virilizationsigns has been highlighted and added in the text (please,see lines 238-241)
Comment 10:testicular examination à look for tumours, look for varicocele and look for spermatic conduct absence… Also include micropenis and clinical suspicion
Answer 10: The role of testicular palpation has been further explored and information onmicropenis has been added (please,see lines 118-153).
Comment 11:breasts exploration à differential diagnosis between gynecomastia and lypomastia, presence of galactorrea or telorrea... might have clinical implications, and should be included
Answer 11: We appreciate the reviewer’s suggestions. Additional information on the differential diagnosis of gynecomastia and the role of galactorrhea has been added (please,see lines 178-191).
Comment 12:Visual fields exploration, for a first impression of vision in a patients with pituitary adenoma with chiasma affectation.
Answer 12: A mention of visual impairment from chiasmatic syndrome has been added in paragraph 6 (please,see lines 421-425).
Comment 13:acromegalic features, key in the diagnostic and for the suspicion of the disease are not detailed (acral enlargement, macroglosia….also in severe hypothyroidism). Physical aspect of a gigantism and a Laron syndrome …
Answer 13: As suggested by the reviewer, the anthropometric parameters section was expanded by including the physical characteristics of acromegaly, gigantism, Laron syndrome, and GH deficiency.
Comment 14:Hypothyroidism: facial edema, macroglosia…
Answer 14: Anthropometric signs of hypothyroidism were added in lines 325-329.
Comment 15:Physical examination of thyroid cyst and thyroglossal duct cyst are not mentioned. Pemberton sign is important for the evaluation of a large endothoracic goiter
Answer 15: We thank the reviewer for the suggestion. Information on the thyroglossal duct cyst and Pemberton’s sign has been added (please,see lines 86-93).
Comment 16:Facial erythema, telangiectasias, might suggest carcinoid syndrome.
Answer 16: As suggested,a mention was made about the physical signs of carcinoid syndrome (please,see lines 281-283)
Minor:
Comment 17: Typo line 38 page 1: semiological instead of semeiological
Answer 17:The mistake was corrected.
Round 2
Reviewer 1 Report
The authors extensively reviewed their manuscript.
However, the manuscripts lacks focus. What is the goal of the authors: a discription of the most outstanding and most common signs, a complete overview of all signs and symptoms or anything in between? Is seems to be a random selection of signs and symptoms, giving the impression of a complete overview while many aspects are lacking.
An example of incomplete presentation: in the paragraph on hyperpigmentation, why are cafe-au-lait maculae, freckling and other hyperpigmentated signs of heriditary syndromes not presented?
Table 1 also gives the suggestion of a very complete overview, while some aspects are incomplete and out of context or even incorrect.
For example in Table 1:
-small testes: this can also be a sign of hypergonadotrophic-hypogonadism, not all boys/men have firm testes with this diagnosis.
- Small firm testes: text indicates hypogonatotropic instead of hypergonadotropic, and Klinefelter is not the only cause.
- high GH and low IGF1: Laron is very rare, while other GH-insensitivity syndromes are much more common.
- Micropenis +high pubic fat deposition is NOT a direct clue for buried penis, this is only the case if the stretched penile length is normal.
In addition, throughout the whole manuscript descriptions are randomly displayed as sign-based in two paragraph (2+3) and disease- of organ based in others (4-6). This is quite confusing.
The manuscript would benefit from a more focused presentation of signs and symptoms, differentiating in common and rare signs and diagnosis.
Author Response
Answers to the comments of Reviewer 1
Manuscript ID: jcm-1633588
Comment 1: The authors extensively reviewed their manuscript. However, the manuscripts lacks focus. What is the goal of the authors: a description of the most outstanding and most common signs, a complete overview of all signs and symptoms or anything in between? Is seems to be a random selection of signs and symptoms, giving the impression of a complete overview while many aspects are lacking.
Answer 1: We agree with the reviewer that other signs could be included to make the overview more complete. However, as also specified in the Introduction, the aim of this article was not to list all the individual signs of endocrine diseases, as in a treatise on semeiotics, but to highlight the main signs that certainly should be sought in the suspicion of endocrine diseases or those signs whose presence allows to orient towards a diagnosis of endocrinological diagnosis. For this reason, we have written this article to highlight only the main signs (please see lines 72-74). In the course of the revision, thanks to the suggestions of the reviewers', the manuscript certainly improved considerably.
Comment 2: An example of incomplete presentation: in the paragraph on hyperpigmentation, why are cafe-au-lait maculae, freckling and other hyperpigmentated signs of heriditary syndromes not presented?
Answer 2: Thank you for this suggestion. We have included a brief mention of cafe-au-lait maculae and lentiginosis in rare genetic syndromes associated with endocrinological abnormalities. We have not particularly elaborated on this section because these are signs that, although often the first manifestation of the disease, are seen later after the onset of other more specific signs (e.g. the appearance of precocious puberty in McCune Albright syndrome).
Comment 3: Table 1 also gives the suggestion of a very complete overview, while some aspects are incomplete and out of context or even incorrect. For example in Table 1:
-small testes: this can also be a sign of hypergonadotrophic-hypogonadism, not all boys/men have firm testes with this diagnosis.
Answer 3: We thank the reviewer for the suggestion. Accordingly, we have deleted hypergonadotropic and left only the term hypogonadism.
Comment 4: - Small firm testes: text indicates hypogonatotropic instead of hypergonadotropic, and Klinefelter is not the only cause.
Answer 4: We have modified the table as suggested. Thank you.
Comment 5: high GH and low IGF1: Laron is very rare, while other GH-insensitivity syndromes are much more common.
Answer 5: We agree with the reviewer that the inclusion of Laron syndrome alone was misleading. Therefore, we have replaced it in both text and table with the term GHIS (growth hormone insensitivity syndromes).
Comment 6: Micropenis + high pubic fat deposition is NOT a direct clue for buried penis, this is only the case if the stretched penile length is normal.
Answer 6: We agree with the reviewer on this point. Therefore, we deleted it from the table and left it only in the text to highlight the importance of the differential diagnosis.
Comment 7: In addition, throughout the whole manuscript descriptions are randomly displayed as sign-based in two paragraph (2+3) and disease- of organ based in others (4-6). This is quite confusing.
Answer 7: We very much appreciate this helpful suggestion. For the sake of greater clarity, we have modified the text. Now, each section begins with the sign and then describes the diagnosis.
Comment 8: The manuscript would benefit from a more focused presentation of signs and symptoms, differentiating in common and rare signs and diagnosis.
Answer 8: We thank the reviewer for her/his suggestion and agree that some signs are more common than others also as reflection of the epidemiology of the various diseases. However, we think that a classification of signs according to their frequency is more complex and requires a new restructuring of the text, differently from what the second reviewer suggested. This would therefore imply a complete change of the main objective of the article which wants to underline the importance of the presence of some signs in guiding in guiding the diagnostic path of the clinician who observes them. However, if the editor finds this reorganization of the text useful, we are willing to do so.
Reviewer 2 Report
Crafa et al, have substantially improved and reorganized the manuscript. I suggest the authors add subcategories in each section to make the reading easy to understand, similar to subcategories (type of sign) included in the table (or by organs, glands, tissues; ie: thyroid, testes...).
I still miss some relevant signs like hyperpigmentation for a primary adrenal insufficiency or in cases of Nelson's syndrome, hypopigmentation (vitiligo) as a sign for the suspicion of autoimmune diseases...,
There are some typos and missing spaces in some words, please double check the spelling.
Author Response
Answers to the comments of Reviewer 2
Manuscript ID: jcm-1633588
Comment 1: Crafa et al, have substantially improved and reorganized the manuscript. I suggest the authors add subcategories in each section to make the reading easy to understand, similar to subcategories (type of sign) included in the table (or by organs, glands, tissues; ie: thyroid, testes...).
Answer 1: We appreciated this comment which helped make our article clearer to the reader. To accomplish this, we have divided the main sections in subsections.
Comment 2: I still miss some relevant signs like hyperpigmentation for a primary adrenal insufficiency or in cases of Nelson's syndrome, hypopigmentation (vitiligo) as a sign for the suspicion of autoimmune diseases...,
Answer 2: We added a subsection to the role of skin pigmentation changes in endocrinology (please, see lines 220-240). The signs have also been added to the table.
Comment 3: There are some typos and missing spaces in some words, please double check the spelling.
Answer 3: The text has been carefully revised.